# Genotyping for HLA Risk Alleles to Prevent Drug Hypersensitivity Reactions: Impact Analysis

**DOI:** 10.3390/ph15010004

**Published:** 2021-12-21

**Authors:** Lisanne E. N. Manson, Wilbert B. van den Hout, Henk-Jan Guchelaar

**Affiliations:** 1Department of Clinical Pharmacy and Toxicology, Leiden University Medical Center, 2333 ZA Leiden, The Netherlands; L.E.N.Manson@lumc.nl; 2Leiden Network for Personalized Therapeutics, 2333 ZA Leiden, The Netherlands; 3Department of Biomedical Data Sciences, Leiden University Medical Center, 2333 ZA Leiden, The Netherlands; W.B.van_den_Hout@lumc.nl

**Keywords:** HLA, drug hypersensitivity, pharmacogenomics, abacavir, allopurinol, flucloxacillin, antiepileptic drugs, cost-effectiveness

## Abstract

Human Leukocyte Antigen (HLA) variants can be a risk factor for developing potentially fatal drug hypersensitivity reactions. Our aim was to estimate the potential impact of genotyping for the HLA risk alleles incorporated in the Dutch Pharmacogenetics Working Group (DPWG) guidelines in The Netherlands. We estimated the number of hypersensitivity reactions and associated deaths that can be avoided annually by genotyping for these HLA risk alleles. Additionally, the cost-effectiveness was estimated. Nationwide implementation of genotyping HLA risk alleles before initiating drugs with an actionable drug–gene interaction can potentially save the life of seven allopurinol initiators and two flucloxacillin initiators each year in The Netherlands. Besides these deaths, 28 cases of abacavir hypersensitivity, 24 cases of allopurinol induced SCARs, 6 cases of carbamazepine induced DRESS and 22 cases of flucloxacillin induced DILI can be prevented. Genotyping *HLA-B*5701* in abacavir initiators has a number needed to genotype of 31 to prevent one case of abacavir hypersensitivity and is cost-saving. Genotyping *HLA-B*5801* in allopurinol initiators has a number needed to genotype of 1149 to prevent one case of SCAR but is still cost-effective. Genotyping before initiating antiepileptic drugs or flucloxacillin is not cost-effective. Our results confirm the need for mandatory testing of *HLA-B*5701* in abacavir initiators, as indicated in the drug label, and show genotyping of *HLA-B**5801 in allopurinol initiators should be considered.

## 1. Introduction

Adverse drug reactions (ADRs) are a major cause of morbidity and mortality in modern healthcare. An adverse drug reaction (ADR) is defined by the World Health Organization as ”a response to a drug which is noxious and unintended, and which occurs at doses normally used in humans for the prophylaxis, diagnosis, or therapy of disease, or for the modifications of physiological function” [1]. Two types of ADRs are distinguished, type A and type B reactions. Whereas type A reactions can be predicted from the drug’s pharmacological mechanism of action, type B reactions, or idiosyncratic reactions, cannot be predicted as such and are often much rarer than type A reactions.

Bouvy et al. performed a review of all epidemiological studies quantifying ADRs in a European setting and found a median percentage of ADR related hospital admissions of 3.5% of all admissions and a fatality rate of approximately 0.15% indicating that ADRs represent a significant burden on European healthcare [2]. Incidence numbers of idiosyncratic ADRs are scarcely available within Caucasian cohorts, but according to Asian studies the incidence of drug hypersensitivity of hospitalized patients is estimated to be 1.8–4.2 per 1000 hospital admissions [3,4]. They are rare but, due to them being unpredictable and often severe with high mortality rates, genetic biomarkers to identify patients at higher risk of drug hypersensitivity are of utmost importance in preventing these hypersensitivity reactions, hospital admissions and potentially death.

In 2005, the Dutch Pharmacogenetics Working Group (DPWG) was established by the Royal Dutch Pharmacist’s Association with the objective to develop pharmacogenetics-based therapeutic (dose) recommendations [5]. Many of its guidelines concern pharmacokinetic and pharmacodynamic gene-drug interactions to prevent type A ADRs. But also guidelines related to risk alleles in the HLA gene and associated idiosyncratic drug reactions are available.

The guidelines include *HLA-B*5701* and abacavir hypersensitivity reaction (ABC-HSR), *HLA-B*5801* and allopurinol associated severe cutaneous adverse drug reactions (SCARs) including Stevens–Johnson syndrome (SJS), toxic epidermal necrolysis (TEN) and drug reaction with eosinophilia and systemic symptoms (DRESS) and *HLA-B*5701* and flucloxacillin induced liver injury (DILI). Additionally there are guidelines about *HLA-B*1502*, *HLA-B*1511* and *HLA-A*3101* and carbamazepine induced SCARs and *HLA-B*1502* and lamotrigine, oxcarbazepine and phenytoin associated SJS/TEN [6,7,8,9,10,11,12,13].

In many populations, these HLA tests have been proven to be effective in predicting susceptibility for the specific drug hypersensitivity reactions and in a recent paper the diagnostic test criteria including sensitivity and specificity have been described [14]. However, it is yet unclear what the impact of genotyping for these HLA risk alleles would be in a European population.

Genotyping for HLA risk alleles potentially has a high impact for the individual patient and nationwide. Therefore we aim to make a quantitative estimate of the potential impact of genotyping for HLA risk alleles in The Netherlands by estimating the number of drug hypersensitivity reactions that can be avoided annually by switching patients to an alternative drug if they tested positive for the specific HLA risk allele. Additionally we determined how many deaths could be prevented annually by this approach. In addition, we performed a cost-effectiveness study to determine whether testing for HLA risk alleles is cost-effective.

## 2. Results

### 2.1. Nationwide Dispensing Data

An overview of first dispensations obtained from the Dutch Foundation for Pharmaceutical Statistics or Stichting Farmaceutische Kerngetallen (SFK) of abacavir, allopurinol, carbamazepine, flucloxacillin, lamotrigine, oxcarbazepine and phenytoin for the period 1 January–31 December 31, 2019 is shown in Table 1 [15]. First dispensations are defined as a dispensation without a previous dispensation within the prior 12 months [16]. Flucloxacillin is the most represented drug of the seven drugs with almost 19,000 first dispensations per million patients followed by allopurinol with around 1700 first dispensations and carbamazepine with nearly 600 dispensations per million patients.

### 2.2. Frequencies of HLA Variants

The frequency of HLA variants is derived from Dutch populations from the Allele Frequency Net Database and literature [17,18]. The most common HLA risk alleles in The Netherlands are *HLA-B*5701* and *HLA-A*3101*. *HLA-B*1511* was not found in any of the four Dutch populations while *HLA-B*1502* was only found once. The weighted averages of the allele frequencies and calculated carrier frequencies of the five HLA risk alleles mentioned in the DPWG guidelines are shown in Table 2.

### 2.3. Positive Predictive Values

The probability that following a positive HLA test result, the individual will develop the specific drug hypersensitivity when exposed to the drug of interest, is the positive predictive value (PPV). The positive predictive values of the HLA-drug hypersensitivity reactions are shown in Table 3. The PPVs were retrieved from literature and existing guidelines. The *HLA-B*5701* test for abacavir hypersensitivity has by far the highest PPV of approximately 48%. The PPV of *HLA-B*5801* for allopurinol-induced SJS/TEN is also relatively high with 5.5%. On the other hand, all other PPVs are below 1%.

### 2.4. Mortality

The mortality rates of abacavir hypersensitivity reaction (ABC-HSR), SJS/TEN, SJS, DRESS and DILI used in our analysis are shown in Table 4. Abacavir hypersensitivity has a mortality of about 0.07%, DRESS of 2%, DILI of 7.6% while SJS and SJS/TEN have a high mortality of 24% and 34%.

### 2.5. Hypersensitivity Reactions and Deaths Prevented by HLA Genotyping

Table 5 summarizes the estimated annual number of actionable genotypes, HLA associated hypersensitivity reactions and deaths caused by these hypersensitivity reactions in The Netherlands. As is shown in Table 5, *HLA-B*5701* testing, which is already mandatory for abacavir prior to start, is estimated to prevent 28 hypersensitivity reactions per year but less than one death each year. This is due to a high allele carrier frequency and positive predictive value but low mortality rate of 0.07%. This low mortality rate can probably be explained by the fact that abacavir hypersensitivity reaction is more severe after a rechallenge and the drug is usually withdrawn after a first reaction.

*HLA-B*5801* testing of all allopurinol starters and switching to an alternative drug when tested positive would prevent 21 cases of SJS/TEN and 3 cases of DRESS each year and is estimated to prevent 7 deaths each year.

Our results show that for some drugs-gene interactions, genotyping for HLA in the Dutch population would prevent less than one hypersensitivity reaction per year. This includes *HLA-B*1502* genotyping for lamotrigine, oxcarbazepine and phenytoin starters and *HLA-B*1511* testing for carbamazepine starters. *HLA-A*3101* testing for carbamazepine starters and switching to an alternative drug when tested positive would prevent 6 cases of DRESS per year.

*HLA-B*5701* testing for all flucloxacillin starters is estimated to prevent 22 hypersensitivity reactions and 2 deaths per year, but almost 300,000 people would need to be genotyped for *HLA-B*5701* each year.

### 2.6. Cost-Effectiveness Model

The net annual costs for the Dutch population for the genotyping strategy versus the standard care was calculated per drug–gene interaction. We considered a cost of 2 million euros per prevented death to be cost-effective [30]. Our results in Table 6 show that *HLA-B*5701* testing for abacavir starters is calculated to be cost saving. Annually, genotyping all 873 abacavir initiators and switching 58 *HLA-B*5701* positive patients to an alternative drug saves €34,000 in net costs in the Netherlands. Although not cost saving like testing for abacavir, genotyping *HLA-B*5801* for allopurinol starters would still be cost-effective. Annually, this would cost about 2 million euros and prevents 24 cases of allopurinol induced SCARS and 7 deaths. Ignoring the prevented ADR, the corresponding cost-effectiveness ratio is €282,000 to prevent one death. This is considered very cost-effective. Genotyping for HLA risk alleles for starters of antiepileptic drugs and flucloxacillin is not cost-effective. *HLA-B*1502*, *HLA-B*1511* and *HLA-A*3101* genotyping for carbamazepine initiators comes closest to being cost-effective with costs of 5 million euros per prevented death.

## 3. Discussion

Nationwide implementation of genotyping for specific HLA risk alleles before initiating drugs with an actionable drug–gene interaction and switching when tested positive can potentially save the life of 7 allopurinol initiators and 2 flucloxacillin initiators each year in The Netherlands. Besides these deaths, 28 cases of abacavir hypersensitivity, 24 cases of allopurinol induced SCARs, 6 cases of carbamazepine induced DRESS and 22 cases of flucloxacillin induced DILI can be prevented. Our analysis confirms the importance of mandatory testing of *HLA-B*5701* for abacavir starters since the number needed to genotype is only 31. Indeed, only 873 patients have to be genotyped and of these 58 *HLA-B*5701* positive patients have to switch to an alternative therapy to prevent 28 cases of abacavir hypersensitivity. This approach saves The Netherlands the limited amount of 34,000 euros each year. Promising seems to be *HLA-B*5801* testing for allopurinol starters with a number needed to genotype of 1149 initiators to prevent one case of SJS/TEN or DRESS. Testing of 27,585 allopurinol starters would prevent 21 cases of SJS/TEN, 3 cases of DRESS and 7 deaths each year. Especially patients with chronic renal insufficiency have a high chance of developing SJS/TEN when exposed to allopurinol. In this patient group, the PPV is tenfold higher. Therefore, genotyping for *HLA-B*5801* of this specific patient group would be the most obvious step forward for HLA testing in the Dutch population. HLA genotyping for AEDs and flucloxacillin is not cost-effective.

To our knowledge, this study is the first nationwide impact analysis and cost-effectiveness study of all, according to the CPIC and DPWG guidelines, actionable HLA-drug interactions. The analysis includes abacavir, allopurinol, carbamazepine, oxcarbazepine, lamotrigine, phenytoin and flucloxacillin. Zhou et al. have developed a global cost-effectiveness model to estimate country-specific cost-effectiveness thresholds for pre-emptive genetic testing for abacavir, allopurinol and carbamazepine [31]. However, in this study, The Netherlands was only included in the cost-effectiveness analysis of carbamazepine. Our study shows that testing for *HLA-B*1502* and *HLA-A*3101* is not cost-effective in the Dutch population. This is in disagreement with Zhou at al who estimated that HLA-A*3101 testing for carbamazepine is likely to be cost-effective, although the carrier frequency used in both studies is comparable. However, they assumed the costs of HLA testing would be only USD 40, which we currently consider too low.

Although our study focusses on the impact and cost-effectivity of genotyping for HLA risk alleles in The Netherlands, the results may be generalizable to other Caucasian European populations, if carrier frequencies and healthcare costs in those populations are comparable to the Dutch values.

A limitation of our study is that we used mainly input data of Caucasian populations. This leads to underestimating the usefulness of *HLA-B*1502* and *HLA-B*1511* for patients with Asian ancestry in The Netherlands for which HLA-B*1502 and HLA-B*1511 genotyping may be more useful than for the average Dutch population. However, because only 5.5% of the Dutch population has migrated from Asia, and of this 5.5% the majority comes from Indonesia where HLA studies are relatively scarce, this was considered to be the best approach. In contrast, we may have overestimated the usefulness of *HLA-B*1502* and *HLA-B*1511* for Caucasian lamotrigine, oxcarbazepine and phenytoin initiators. No data for a PPV in Caucasian populations was found for these drug–gene interactions and therefore we used PPVs originating from Asian studies which may have led to an overestimation of the utility of the HLA tests.

For an estimation of the number of patients initiating abacavir, allopurinol, flucloxacillin or one of the four AEDs we used data of first dispensations from the Dutch Foundation for Pharmaceutical Statistics. Conventionally, first dispensations are defined as a pharmacy dispensation without a previous dispensation of the same drug in the prior 12 months in the community pharmacy [16]. This probably leads to an overestimation of actual first starters, especially for flucloxacillin. The other drugs are often used chronically for many years while flucloxacillin can be used for a short duration multiple times in a lifetime but with more than 12 months between consecutive uses.

For the costs of treating the drug hypersensitivity reactions, we used the average of the prices charged by the Dutch academic hospitals. To check whether these are representative, we compared the used to costs found in the literature. The price used for abacavir hypersensitivity reaction was comparable to the costs of treating abacavir hypersensitivity reaction in a UK study and a German study [32,33]. The costs for treating DRESS used in this study (6800 euros) was considerably lower than found in a US study (17,000 USD) [34]. This is consistent with the generally higher price level of US healthcare, as compared to other high-income countries [35]. The average price for Dutch academic hospitals we used for SJS/TEN (9100 euros) was considerably lower than in a Dutch study in a burn center specialized in SJS/TEN [36]. We attribute this to the longer treatment duration and especially ICU stay for the severe cases that are referred to this burn center, as compared to the SJS/TEN in our study. Overall, we can conclude that the prices used in our cost-effectiveness model are comparable to costs in other western countries and can therefore be considered reliable estimates.

In this study we only focused on HLA associated hypersensitivity reactions and we did not take into account any other ADR of the studied drugs or of the drugs that could be used in case of a hypersensitivity reaction. Obviously, if a patient is tested positive for an HLA risk allele and the physician prescribes an alternative drug, this drug may also cause ADRs, or it might be less effective. Febuxostat, for instance, seems to have more cardiovascular adverse drug reactions than allopurinol but can also cause cutaneous hypersensitivity reactions, though these are rarer than with allopurinol and not associated with an HLA risk allele or other predictive biomarker [37,38,39].

We performed a cost-effectiveness analysis where we calculated genotyping for abacavir to be cost-saving, and genotyping for allopurinol to be cost-saving while for the other drugs genotyping is not cost-effective. We used a threshold of 2 million euros per prevented death, although the literature is not very clear what the threshold should be [30]. In many countries economic thresholds in terms of costs-per-QALY are used [40]. However, we felt using QALYs for our calculations would require making many uncertain assumptions and exploratory calculations suggested it would not change our conclusion.

Currently, the implementation of HLA testing in patients starting drugs with a known gene-drug interaction in The Netherlands remains low, despite available guidelines and actionability. Only for abacavir initiators, genotyping for *HLA-B*5701* is mandatory and routinely done and our study confirms this approach is cost-saving. The other HLA alleles are rarely tested before initiating drug therapy. Our study shows genotyping *HLA-B*5801* for allopurinol starters is cost-effective and can reduce morbidity and mortality and therefore it should be considered in pharmacogenetic implementation programs. Also genotyping carbamazepine starters with Asian ancestry should be considered, because of the high prevalence of *HLA-B*1502* in this population. However, genotyping all carbamazepine starters, regardless of ethnicity, is not cost-effective. Also genotyping for flucloxacillin, oxcarbazepine, lamotrigine and phenytoin is not cost-effective in the Dutch population.

## 4. Materials and Methods

### 4.1. Selection of Drug Gene Interactions

Gene-drug interactions with an actionable therapeutic recommendation for carriers of HLA risk alleles were selected from the DPWG guidelines. These include abacavir-*HLA-B*5701*, allopurinol-*HLA-B*5801*, carbamazepine-*HLA-B*1502*, *A*3101* and *B*1502*, flucloxacillin-*HLA-B*5701*, lamotrigine-*HLA-B*1502*, oxcarbazepine-*HLA-B*1502* and phenytoin-*HLA-B*1502.* The drug hypersensitivity reactions in this article are the drug hypersensitivity reactions as described in the DPWG guidelines [7,8,9,10,11,12,13].

### 4.2. Source of Nationwide Dispensing Data

Dutch dispensing data from “first dispensations” of the 7 selected drugs were collected for the period 1 January–31 December 2019 from the Dutch Foundation for Pharmaceutical Statistics or Stichting Farmaceutische Kengetallen (SFK). SFK is the Dutch Foundation for Pharmaceutical Statistics and collects exhaustive data about the use of dispensed drugs in the Netherlands. It contains data from more than 98% of the approximately 2000 community pharmacies in the Netherlands. These pharmacies combined serve about 15.8 million people from the population in The Netherlands [15]. Conventionally, first dispensations are defined as a dispensation without a previous dispensation within the prior 12 months in the community pharmacy [16].

### 4.3. Frequencies of HLA Variants

The allele frequencies of the HLA variants in the Dutch population were collected from the Allele Frequency Net Database and from a PubMed search. In the Allele Frequency Net Database all populations concerning the country The Netherlands were included. A PubMed search was performed using the search terms “(hla [ti]) AND (dutch [ti] OR Netherlands [ti])” and relevant articles which include allele frequencies of the HLA variants of interest were included.

Three Dutch cohorts were found in the Allele Frequency Net Database and one cohort was retrieved using the PubMed search [17,18]. A weighted average of the allele frequencies of the 4 populations was calculated by performing a meta-analysis in R. Because of low heterogeneity, a fixed effects model was used.

The carrier frequency was subsequently calculated as follows:Carrier frequency = 1 − (1 − allele frequency)^2^.

### 4.4. Positive Predictive Values

The PPVs for the actionable drug–gene interactions were retrieved from studies performed in Caucasian populations or from the DPWG guideline. If available, numbers for Caucasian populations were used, otherwise PPVs derived from other populations were used.

### 4.5. Morbidity and Mortality of Outcomes

The hypersensitivity reactions included in this article are abacavir hypersensitivity, allopurinol induced SCARs including DRESS and SJS/TEN, carbamazepine induced SCAR, lamotrigine and phenytoin induced SJS/TEN and oxcarbazepine induced SJS. Finally flucloxacillin induced DILI was also included. Estimates of the mortality of the included drug hypersensitivity reactions were collected from literature [26,27,28,29]. If possible, mortality estimates from European or other mainly Caucasian countries were used.

### 4.6. Estimation of Hypersensitivity Reactions and Deaths Prevented by Pre-Emptively Genotyping

The number of actionable drug–gene interactions is calculated by multiplying the number of first prescriptions by the allele carrier frequency of the associated HLA risk allele. Subsequently the estimated annual number of drug hypersensitivity reactions is calculated by multiplying the number of actionable drug–gene interactions by the positive predictive value of the HLA test for the outcome. Thus the annual number of drug hypersensitivity reactions is calculated by:(1)$$N_{\mathrm{ADR}}=\;\mathrm{DISP}\times \mathrm{CF}\times \mathrm{PP}V$$

N_ADR_ = number HLA associated hypersensitivity reactions prevented, DISP = number of first dispensations, CF = carrier frequency of HLA risk allele, PPV = positive predictive value of HLA test.

The annual number of deaths caused by HLA associated hypersensitivity reactions is calculated by multiplying the estimated number of drug hypersensitivity reactions by the mortality of the outcome:(2)$$N_{\mathrm{death}}=N_{\mathrm{ADR}}\times \mathrm{MORT}$$

N_death_ = number of HLA associated deaths prevented, N_ADR_ = number HLA associated hypersensitivity reactions prevented, MORT = mortality.

### 4.7. Healthcare Costs

We compared the cost of two treatment strategies; no genetic test and initiation of the HLA allele associated drug, and testing for the HLA risk allele and switching to an alternative drug not associated with the same HLA risk allele when a patient tests positive for the HLA risk allele.

Costs are calculated with a one-year time-horizon, and reported in Euros at price level 2021. The costs of single-variant HLA tests were based on single-gene prices for genotyping (DNA amplification, manually) and a fixed price per lab-order for handling and administration set in the four hospitals who test for *HLA-B*5701* separately. The cost of drugs for the relevant reference drugs and the alternatives was based on the most common indication for the drug. The costs of the reference and alternative drugs were collected from the national drug price registry by selecting the most suitable dose and formulation [41]. The alternative drugs were chosen from existing guidelines for the most common indication of the drugs of interest.

In the case of flucloxacillin, unlike the other drugs of interest, the DPWG guideline does not recommend an alternative drug but instead recommends to monitor liver function. Only with increasing liver values, an alternative drug is recommended [8]. For the costs of the monitoring, the average cost from the Dutch academic hospitals was used for testing ALAT and ASAT and also includes a fixed price per lab-order for handling and administration and blood withdrawal. According to previous research, 0.0097% of all flucloxacillin initiators patients experience increasing liver values [42].

The costs used in the cost-effectiveness model are shown in Table 7.

For the costs of treating drug hypersensitivity reactions the average of the prices charged by the seven academic hospitals (Amsterdam University Medical Center, University Medical Center Groningen, Leiden University Medical Center, Maastricht University Medical Center+, Radboud University Medical Center, Erasmus University Medical Center and University Medical Center Utrecht in The Netherlands was used. However, as a check to see whether these costs are representative a literature search was performed to compare the prices from the academic hospitals to the costs in previous clinical studies.

For abacavir hypersensitivity the price for “intensive treatment for allergy” was used (3700 euros). The European RegiSCAR study found a median stay of 17 days for DRESS patients. Because of this, the price for “6 to 28 days of treatment for inflammation of the skin” was used (6800 euros). For SJS, SJS/TEN and TEN the price for “6 to 28 days of treatment for a skin condition with blisters” was used (9100 euros). The costs used for DILI are an average of the prices for patient receiving a liver transplantation after suffering from DILI and patients not needing a liver transplantation. In the DILIN prospective study 3.8% of the patients suffering from DILI needed a liver transplant [29]. The price for “liver transplantation with a living donor with a maximum of 28 days due to severe liver failure” was used for patients needing a liver transplantation while the price for “hospital admission of 6–28 days for a disease of the liver” was used for the remaining 96.2% not needing a liver transplant.

### 4.8. Cost-Effectiveness Model

The net annual costs for the Dutch population for the genotyping strategy versus the standard care was calculated per drug–gene interaction as following:(3)$$\mathrm{DISP}\times C_{\mathrm{test}}+{\mathrm{HLA}}^{+}\times C_{\mathrm{mon}}+\left({\mathrm{HLA}}^{+}-\mathrm{ADR}\right)\times 365\times \left(C_{\mathrm{alt}}-{\;C}_{\mathrm{ref}}\right)-\mathrm{ADR}\times C_{\mathrm{ADR}}$$

With DISP as the number of first dispersions, C_test_ as the costs of the HLA test, HLA+ as the number of patients carrying the risk allele, C_mon_ as the costs for monitoring liver values in the case of flucloxacillin, ADR as the number of patients developing drug hypersensitivity, C_alt_ and C_ref_ as the daily drug costs of respectively the alternative and reference drug and C_ADR_ as the estimated costs of an hypersensitivity reaction.

We calculated cost-effectiveness ratios by dividing this impact on costs by the impact on the number of ADRs and deaths.

## 5. Conclusions

This study confirms the need for mandatory testing of *HLA-B*5701* in abacavir initiators, as indicated in the drug label, since it only has a number needed to genotype of 31 and implementation of this test is calculated to be cost-saving. Genotyping all carbamazepine, oxcarbazepine, lamotrigine, phenytoin and flucloxacillin initiators is not cost-effective in the overall Dutch population. However, our study shows genotyping *HLA-B*5801* for allopurinol starters is cost-effective and can reduce morbidity and mortality and therefore it should be considered in pharmacogenetic implementation programs.

## Figures and Tables

**Table 1 pharmaceuticals-15-00004-t001:** An overview of the first dispensations for drugs with an actionable HLA associated DPWG recommendation dispensed in Dutch pharmacies [15].

Name Drug	No. First Dispensations	First Dispensations per Million Patients
FLUCLOXACILLIN	296,467	18,529
ALLOPURINOL	27,585	1724
CARBAMAZEPINE	9520	595
LAMOTRIGINE	6426	402
OXCARBAZEPINE	1219	76
ABACAVIR monotherapy or in combination with other antiviral drugs	873	55
PHENYTOIN	678	42

**Table 2 pharmaceuticals-15-00004-t002:** Allele frequencies and carrier frequencies of HLA risk alleles in the Dutch population [17,18].

HLA Risk Allele	Allele Frequency	Carrier Frequency
*HLA-A*3101*	0.0338	0.0665
*HLA-B*1502*	0.0007	0.0014
*HLA-B*1511*	0.0000	0.0000
*HLA-B*5701*	0.0336	0.0661
*HLA-B*5801*	0.0069	0.0138

**Table 3 pharmaceuticals-15-00004-t003:** Positive predictive values of HLA-drug hypersensitivity interactions.

Drug	HLA Variant	Outcome	PPV	Source
Abacavir	*HLA-B*5701*	Abacavir hypersensitivity reaction	48%	DPWG guideline [12]
Allopurinol	*HLA-B*5801*	SJS/TEN DRESS	5.5% 0.72%	Lonjou et al. [19] Gonçalo et al. [20]
Flucloxacillin	*HLA-B*5701*	DILI	0.11%	Daly et al. [21]
Carbamazepine	*HLA-B*1502*	SJS/TEN	0.14%	Amstutz et al. [22]
	*HLA-B*1511*	SJS/TEN	0.5%	DPWG guideline [7]
	*HLA-A*3101*	DRESS SJS/TEN	0.89% 0.02%	Genin et al. [23] DPWG guideline [7]
Oxcarbazepine	*HLA-B*1502*	SJS	0.73%	Chen et al. [24] DPWG guideline [10]
Lamotrigine	*HLA-B*1502*	SJS/TEN	0.4%	DPWG guideline [9]
Phenytoin	*HLA-B*1502*	SJS/TEN	0.65%	Chen et al. [25] DPWG guideline [11]

**Table 4 pharmaceuticals-15-00004-t004:** Mortality rates of hypersensitivity reactions.

Type of Hypersensitivity Reaction	Mortality (Reference)
ABC-HSR	0.07% [26]
DRESS	2% [27]
SJS	24% [28]
SJS/TEN	34% [28]
DILI	7.6% [29]

**Table 5 pharmaceuticals-15-00004-t005:** An overview of actionable genotypes amongst drug initiators and the estimated number of hypersensitivity reactions and deaths caused by hypersensitivity reactions that can be prevented by HLA genotyping.

Drug	HLA Allele	Outcome	First Prescriptions	Carrier Frequency	PPV	Mortality	Actionable Genotypes	Hypersensitivity Reactions	Deaths
Abacavir	*B*5701*	ABC-HSR	873	0.0661	0.48	0.0007	57.71	27.70	0.02
Allopurinol	*B*5801*	SJS/TEN	27,585	0.0138	0.055	0.34	380.67	20.94	7.12
Allopurinol	*B*5801*	DRESS	27,585	0.0138	0.007	0.02	380.67	2.74	0.05
Carbamazepine	*B*1502*	SJS/TEN	9520	0.0014	0.0014	0.34	13.33	0.02	0.01
Carbamazepine	*B*1511*	SJS/TEN	9520	0	0.005	0.34	0.00	0.00	0.00
Carbamazepine	*A*3101*	DRESS	9520	0.0665	0.0089	0.02	633.08	5.63	0.11
Carbamazepine	*A*3101*	SJS/TEN	9520	0.0665	0.0002	0.34	633.08	0.13	0.04
Flucloxacillin	*B*5701*	DILI	296,467	0.0661	0.0011	0.076	19,596.47	12.56	1.64
Lamotrigine	*B*1502*	SJS/TEN	6426	0.0014	0.004	0.34	9.00	0.04	0.01
Oxcarbazepine	*B*1502*	SJS	1219	0.0014	0.0073	0.24	1.71	0.01	0.00
Phenytoin	*B*1502*	SJS/TEN	678	0.0014	0.0065	0.34	0.95	0.01	0.00

**Table 6 pharmaceuticals-15-00004-t006:** Cost-effectiveness model parameters and results (all in euros).

Drug	HLA Allele	Outcome	Costs HLA Test	Costs Lab	Standard Costs per Day	Alternative Medsper Day	Costs per ADR	Net Costs NL	Net Costs per Patient	Costs per Prevented ADR	Costs per Prevented Death
abacavir	*B*5701*	ABC-HSR	79		29.62	29.54	3700	−34,000	−39	−1200	−1,770,000
allopurinol	*B*5801*	SJS/TEN	79		0.20	0.60	9100	2,040,000	74	97,000	287,000
allopurinol	*B*5801*	DRESS	79		0.20	0.60	6800	2,220,000	80	808,000	40,400,000
cllopurinol total			79		0.20	0.60		2,020,000	73	85,000	282,000
carbamazepine	*B*1502*	SJS/TEN	79		0.45	0.86	9100	754,000	79	41,400,000	119,000,000
carbamazepine	*B*1511*	SJS/TEN	79		0.45	0.86	9100	752,000	79	-	-
carbamazepine	*A*3101*	DRESS	79		0.45	0.86	6800	808,000	83	143,000	7,170,000
carbamazepine	*A*3101*	SJS/TEN	79		0.45	0.86	9100	846,000	89	6,680,000	19,600,000
carbamazepine total			79		0.45	0.86		808,000	83	143,000	4,990,000
flucloxacillin	*B*5701*	DILI	79	19	0.62	0.62	13,500	23,500,000	79	1,090,000	14,300,000
lamotrigine	*B*1502*	SJS/TEN	79		0.36	0.86	9100	509,000	79	14,100,000	41,600,000
oxcarbazepine	*B*1502*	SJS	79		0.87	0.86	9100	96,000	79	7,700,000	22,700,000
phenytoin	*B*1502*	SJS/TEN	79		0.21	0.86	9100	54,000	79	8,700,000	25,600,000

**Table 7 pharmaceuticals-15-00004-t007:** Model inputs for the healthcare costs.

Input	Costs in Euros	Source
*HLA-A*3101* test	79	Average of 4 Dutch hospitals
*HLA-B*1502* test	79	Average of 4 Dutch hospitals
*HLA-B*1511* test	79	Average of 4 Dutch hospitals
*HLA-B*5701* test	79	Average of 4 Dutch hospitals
*HLA-B*5801* test	79	Average of 4 Dutch hospitals
Liver monitoring for HLA-B*5701 positive flucloxacillin users	19	Average of price charged by the Dutch academic hospitals
Triumeq (dolutegravir/abacavir/lamivudine)	29.62 per day	Medicijnkosten.nl [41]
Biktarvy (bictegravir/emtricabine/tenofoviralafenamide)	29.54 per day	Medicijnkosten.nl [41]
Allopurinol	0.20 per day	Medicijnkosten.nl [41]
Febuxostat	0.62 per day	Medicijnkosten.nl [41]
Lamotrigine	0.36 per day	Medicijnkosten.nl [41]
Phenytoin	0.21 per day	Medicijnkosten.nl [41]
Oxcarbazepine	0.87 per day	Medicijnkosten.nl [41]
Carbamazepine	0.45 per day	Medicijnkosten.nl [41]
Levetiracetam	0.86 per day	Medicijnkosten.nl [41]
Flucloxacillin	0.62 per day	Medicijnkosten.nl [41]
Clarithromycin	0.48 per day	Medicijnkosten.nl [41]
Abacavir hypersensitivity reaction	3700	Average of price charged by the Dutch academic hospitals
DRESS	6800	Average of price charged by the Dutch academic hospitals
SJS/TEN	9100	Average of price charged by the Dutch academic hospitals
SJS	9100	Average of price charged by the Dutch academic hospitals
DILI	13,500	Average of price charged by the Dutch academic hospitals

## Data Availability

Data is contained within the article.

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
