# Peer review of "Genotyping for HLA Risk Alleles to Prevent Drug Hypersensitivity Reactions: Impact Analysis"

_pharmaceuticals, 2021, doi:10.3390/ph15010004_

Round 1

Reviewer 1 Report

This is a great and important study by Manson et al. where the authors estimated the number of hypersensitivity reactions and associated deaths that can be avoided annually by genotyping for these HLA risk alleles. In addition, they also estimated the cost-effectiveness of such implementation. This is a well-designed study and the conclusions drawn from the results are appropriate.

I have some minor comments for the authors to address:

[1] “For the costs of treating drug hypersensitivity reactions the average of the prices charged by the eight academic hospitals in The Netherlands was used. However, as a check to see whether these costs are representative a literature search was performed to compare the prices from the academic hospitals to the costs in previous clinical studies.”

- It would be great if the authors elaborate a bit about this, e.g., what are the Hospitals and whether they found a difference from previous clinical studies.

[2] Can the authors elaborate on why their study results do not agree with Zhou et al results of cost-effectiveness for HLA-A*3101/Carbamazepine? What are the main differences in endpoints?

Reviewer 2 Report

It is a very interesting and well written manuscript about the relevance of genotyping HLA risk alleles to prevent drug hypersensitivity reactions, that should be useful for the health and scientific community.

I suggest using italics for HLA alleles.

Reviewer 3 Report

The manuscript “Genotyping for HLA risk alleles to prevent drug hypersensitivity reactions: impact analysis” deals with an important pharmacogenetic aspect represented by potential serious hypersensitivity reactions deriving from the administration of 7 selected drugs in the presence of HLA deleterious polymorphisms. The study is based on a one-year data by the Dutch Foundation for Pharmaceutical Statistics that collects information on dispensed drugs by about 2000 pharmacies in the Netherlands. The authors also provided an economic analysis.

Results clearly show the need to perform up-front genotyping in case of abacavir and are suggestive of a similar approach for allopurinol.

The study has been rigorously designed, performed, and written.

Minor comments:

Although, as reported by the authors, this study represents the first nationwide impact analysis and cost-effectiveness study on actionable HLA-drug interactions, a better contextualization at European and/or worldwide level would be appreciated.

Please, check the correctness of the term “dispensions” in Table 1. “Dispensations” could be more correct.

In Table 1, it would be desirable to list drugs according to the decreasing number of first dispensation.
